# Parental Book-Reading to Preterm Born Infants in NICU: The Effects on Language Development in the First Two Years

**DOI:** 10.3390/ijerph182111361

**Published:** 2021-10-29

**Authors:** Erica Neri, Leonardo De Pascalis, Francesca Agostini, Federica Genova, Augusto Biasini, Marcello Stella, Elena Trombini

**Affiliations:** 1Department of Psychology “Renzo Canestrari”, University of Bologna, 40127 Bologna, Italy; leonardo.depascalis@unibo.it (L.D.P.); federica.genova@unibo.it (F.G.); elena.trombini@unibo.it (E.T.); 2Donor Human Milk Bank Italian Association (AIBLUD), 20126 Milan, Italy; augustoclimb@gmail.com; 3Pediatric and Neonatal Intensive Care Unit, Maurizio Bufalini Hospital, 47521 Cesena, Italy; marcello.stella@auslromagna.it

**Keywords:** book reading intervention, language development, preterm infants, VLBW, longitudinal study, NICU

## Abstract

Background: After preterm birth, infants are at high risk for delays in language development. A promising intervention to reduce this risk is represented by the exposure to parental voices through book-reading in Neonatal Intensive Care Units (NICU). This study investigated the possible advantages of book-reading to preterm neonates during their NICU stay on their subsequent language development. Methods: 100 families of preterm infants were recruited. The parents of 55 preterm infants (Reading Group) received a colored picture-book on NICU admission and were supported to read to their neonate as often as possible and to continue after hospital discharge. Forty-five infants (Control Group) were recruited before the beginning of the intervention. Infant language development was assessed with the Hearing and Language quotients of the Griffith Mental Development Scale at the corrected ages of 3, 6, 9, 12, 18 and 24 months. Results: Regardless of group membership, Hearing and Language mean quotients decreased between 9 and 18 months; nevertheless, this decrease was considerably reduced in the Reading group, compared to the Control Group. Conclusions: Reading in NICUs represents a suitable intervention that could positively influence language development and parent-infant relationships in preterm children. The study findings support its implementation as a preventive measure.

## 1. Introduction

Preterm birth is defined as a birth that occurs before the end of 37th gestational week [1] and it represents one of the leading causes of infant mortality and disability [2,3]. The consequences of premature birth have been found to persist in the long-term, and negatively influence several areas of child development [4,5,6,7]. In particular, previous literature has underlined impairments in language development [8,9,10] where, compared to full-term infants, those born preterm showed decreased word production [11,12], shorter utterances [13,14], and weaker grammatical [15,16] and reading skills [17]. While these delays were mainly observed in school-aged individuals, early difficulties may be found during the first months of life, such as lower production of syllables, imitation of adults’ verbalizations, and discrimination between native and non-native stimuli [11,18,19]. These difficulties may be further exacerbated by particularly severe prematurity, as in the case of infants born with a birth weight under 1500 g (Very Low Birth weight-VLBW; [8,20]). 

The roots of the aforementioned impaired development could be found in both fetus immaturity at birth and in the stress experienced in the Neonatal Intensive Care Unit (NICU). Indeed, preterm birth occurs during a critical moment for fetal cerebral maturation and the development of the auditory system [21]. At 23–24 weeks of gestational age, human fetuses are able to remember word sounds [22] and extract prosodic properties of speech; they can discriminate all phonetic contrasts of languages, recognizing the sound of a familiar word [23]. Full maturation of these abilities in preterm infants may not occur in the womb, but in the atypical early-life environment provided by the NICU, where they are exposed to medical treatment and pain stressors [24]. In particular, in NICUs, infants hear a variety of direct, unpredictable, and disturbing noises caused by, for example, monitor alarms and medical equipment. Conversely, other sensory experiences of prenatal life, such as the exposition to rhythmical, coherent, familiar, and indirect sounds, like maternal heartbeat sounds and voice [25,26,27], are not present. This absence may prove detrimental, because the maternal voice represents an important stimulus for the development of the fetus’ auditory system and later healthy language development [28,29]. 

For these reasons, after preterm birth, the Newborn Individualized Developmental Care and Assessment Program (NIDCAP) provides guidelines for structuring an appropriate physical environment in the NICU for infants and families to continue down the normal developmental path started in the womb. In particular, NIDCAP gives practical routines that should protect infants from unpleasant experiences, such as loud noises, that may overload and stress the infant’s system [30,31]. Furthermore, NIDCAP encourages parents to be part of their infant’s care and aims for parents to be supported to provide preterm infants with appropriate kinds and amounts of stimulation, such as stroking caressing, and talking. In this context, providing preterm infants with the possibility of hearing their parents’ voices may prove to be of particular importance. 

Several studies have investigated the effects of exposure to taped maternal voice sounds [21,32,33,34,35,36,37,38,39,40,41,42]. These studies showed improvement in feeding [35,39], and neuromotor [21,41] and cognitive development. In particular, Caskey [36,37] observed more conversational turns and expressive language at 18 months in preterm infants exposed to maternal voice sounds, compared to preterm infants without this exposure. Nevertheless, other authors found low or no effects of this exposure, especially in relation to physiological outcomes [32,33,38,40,41,43]. Furthermore, a recent study by Lejeune and colleagues [44] found negative outcomes, such as worse tactile sensory learning, for preterm infants exposed to taped maternal voice sounds.

An alternative line of studies has focused on exposure to live maternal voice [45,46,47,48]. This approach was found to bring about significant improvements in physiological outcomes [25,26,27,49]. To our knowledge, however, no study has investigated its effects on later language development. Nevertheless, this approach may be particularly useful in the context of developmental care, because the “direct” live exposure to the maternal voice implies the parent’s physical presence and its direct involvement in the care of the premature infant. The use of live experiences of parental voice also responds to the infant’s need to be engaged in social interactions with adult caregivers from the first days of life, and, specifically, to receive contingent and synchronous responses to their own signals and cues [50,51,52,53]. Reliance on pre-recorded maternal voice sounds would, naturally, neither include nor promote these parent-infant reciprocal interactions. 

Nevertheless, a potential obstacle to this “live” approach can be found in the difficulties that parents of preterm infants may have in engaging spontaneously with their infants, especially considering the impact of the traumatic experience of preterm birth [54,55]. Indeed, these parents may often show some difficulties in talking to preterm infants in their incubators, given their vulnerabilities and very weak interactive skills [56]. 

In consideration of both the benefits of exposing preterm infants to parental voices to support and enhance infant development, and of the difficulties parents may face in verbally engaging their infants, a potentially promising way to channel interventions could be found in the act of book-reading. This kind of intervention would give parents a more structured context for interacting with their infants, compared to completely free talk. Thanks to the pictures and the narrative contents of the books, parents could initiate, support and encourage the acquisition of words; moreover, the pictures included in the book could stimulate parents to label objects, and elaborate on the contents of the book, for example through questions and comments [57,58].

Book-reading has been extensively investigated in wider full-term infant populations, finding positive effects on parent-infant relationship [59,60], on functional brain connectivity [61,62], and especially on language development [63,64,65,66,67,68]. Positive benefits of book-reading interventions on linguistic skills were also observed in samples of preterm infants [69,70], although these studies assessed interventions implemented in the second year of infant life and did not consider the effects of early infant exposure to book-reading. A separate line of studies investigated the feasibility of book-reading intervention in the context of the NICU, finding that it is well-accepted by parents and not intrusive [56,71,72]; moreover, it was found to be associated with improvements in parental knowledge about when and how to read to their infant [73], and in their attitudes towards this reading practice [74]. Nevertheless, these studies did not assess possible effects on infant language skills. 

Given these premises, the objective of the present study was to evaluate the influence of a book-reading intervention for preterm infants during their stay in the NICU on their later language development, as assessed in the first two years of infant corrected age, which represent a crucial period for infant development. We hypothesized that this reading intervention would be associated with increased language skills, controlling for the effects of infant and maternal variables (infant gender, birth weight, gestational age, and maternal education).

## 2. Materials and Methods

### 2.1. Participants

This study was conducted at the NICU of the Bufalini Hospital (Ausl Romagna, Cesena, Italy) and at the Laboratory of Developmental Psychodynamics (Department of Psychology, University of Bologna, Cesena, Italy). All preterm infants admitted in the NICU during the period between March 2011 and December 2013 were included in the Reading Group (RG). Their scores were compared with those of a Control Group-CG, composed of preterm infants hospitalized in the same NICU before intervention implementation (November 2008–February 2011). 

Inclusion criteria for both groups were: infant birthweight below 1500 g (VLBW infants); a length of stay in the NICU of at least five days; and absence of chromosomal abnormalities, cerebral palsy, malformations, and fetopathy. Birth weight was chosen as criterion for inclusion as it is widely used in the Italian context (where the present study was conducted) for the premature infant inclusion and participation in clinical follow up assessments [75,76].

During the study period, 115 preterm infants were considered eligible for the study. Among these, the families of 15 infants did not complete all study assessments, due to scheduling conflicts, leading to a final sample of 100 preterm infants. 

### 2.2. Procedure and Measures

Mothers and fathers were recruited to the study by the medical staff of the NICU involved. NIDCAPs Programs followed in the two periods the two groups were recruited were the same, except for the book-reading interventions, and were based on Developmental Care, i.e., methods aimed to adjust the NICU environment to diminish stress, and support infant behavioral organization, a physiological stability, and sleep rhythms, while promoting neural growth and maturation [77].

Regarding the RG, the book-reading intervention consisted in providing a picture book to parents, written in their native language. At the beginning of hospitalization, the head nurse explained to parents the benefits of book-reading and invited them to read the book either at the bedside or at the port of the incubator, or while holding the infant through kangaroo care (a form of skin-to-skin contact that reinforces the physical and supports affective closeness between parent and preterm infant [31]). Parents could choose their book in the bookshelves of the NICUs; all books were previously chosen according to NATI PER LEGGERE guidelines for Italian infants (www.natiperleggere.it accessed on 2 March 2011). Non-Italian-speaking parents could choose a picture book written in their native language. From that moment, the book became a part of the belongings of the specific infant and their family: the first page was personalized with the infant’s name, hand or footprint and the date the reading began. At discharge from hospital, parents were asked to take the books home and were encouraged to continue reading the same book, as an activity to extend to the whole family [71]. 

All of the NICU staff were trained in the book-reading program and in how to support parents in the use of “baby-talk rules”, characterized by simple and repetitive speech, with a warm and exaggerated intonation pattern; this communication was reinforced through smiling and mirroring of facial expressions. According to Developmental Care, parents carefully complied with NICU rules about the noise, which was monitored and visualized by the fit medical equipment (SoundEar^®^) [78], and chose an appropriate time of interaction with their neonates (e.g., when they were not fretful or hungry). Parents also received written information about the book-reading program (written in the English language for non-Italian-speaking parents). 

Regarding the CG, the staff were involved in supporting and enhancing parents’ abilities to recognize and adequately respond to their infant’s cues, according to Developmental Care; however, no recommendations on reading to infants were given.

After discharge from hospital, all infants were included in a clinical and neurodevelopment follow-up program, which included an assessment of infant development at 3, 6, 9, 12, 18 and 24 months of corrected age. At the first assessment (that is, at three months of infant corrected age), all families were invited to visit the Laboratory of Developmental Psychodynamics where parents completed a written informed consent form, as required by the Italian law (Art. 13 of Law no. 196/2003) and an ad hoc questionnaire regarding socio-demographic variables (e.g., age, nationality, education). Hospital medical records provided the following data: gender; birth weight (Extremely Low Birth Weight-ELBW vs. VLBW; <1000 and <1500 g respectively); and gestational age (Extremely Low Gestational Age-ELGA vs. Very Low Gestational Age-VLGA; <28 and <32 gestational weeks, respectively).

The level of infant development was evaluated at 3, 6, 9, 12, 18, and 24 months of infants corrected age, by a trained psychologist, using Griffiths Mental Development Scales-Revised version (GMDS-R for 0–2 years) [79]. These scales are a widely recognized measure for the assessment of mental and psychomotor development in preterm infants [75,76,80,81]. They assess five main areas: locomotor, personal-social, hearing and language, eye–hand coordination, and performance. The results of the GMDS-R are standardized and relayed in terms of quotients: scores below more than 1 standard deviation below the average are considered at risk of neurodevelopmental impairment. Quotients may be scored separately; in order to exclude the influence of possible confounding variables, only the Hearing and Language Quotient subscale was used for analysis. 

The ethical approval for this study was granted by the CEEIAV Committee (Reg. Sperimentazione n.1587; prot. 2426/2016).

### 2.3. Data Analysis

Demographic variables were compared between groups, using independent sample t tests and Chi Square tests. 

Two-level mixed effects growth curve models were used, using the Hearing and Language Quotient subscale of the GMDS-R as dependent variable, and the main and interactive effects of group (Reading vs. Control) and infant age (in corrected months) as predictors. The main effects of infant gender, birth weight, and gestational age, as well as of maternal education level, were also controlled for in the model, according to previous literature [75,76,82,83,84].

A *p*-value ≤ 0.05 was considered significant, in line with our previous studies. All *p*-values tied to hypothesis testing (i.e., related to the effects of the reading intervention and infant age), in the two models reported, were corrected for multiple testing using the Benjamini–Hochberg FDR method [85].

All analyses were conducted in R(R Foundation for Statistical Computing, Vienna, Austria) [86].

## 3. Results

### 3.1. Sample Characteristics

Sample characteristics are shown in Table 1. The Reading and Control groups included 55 infants and 45 infants, respectively. No group differences were found in relation to infant gender, gestational age, birth weight, maternal age, education level, nationality and marital status.

### 3.2. Language Development across Time

Mean scores for the Hearing and Language Quotient of the GMDS-R are shown in Table 2.

Visual inspection of plotted data for Hearing and Language scores (see Figure 1) suggested a cubic trajectory would best approximate the effect of infant age on these scores.

Thus, a growth curve model was tested, including the linear, quadratic, and cubic effects of infant age and their interactions with the group. Results are shown in Table 3.

The significant main effects of linear, quadratic, and cubic infant age showed that, regardless of group membership, Hearing and Language scores decreased between 9 and 18 months of corrected age, with greater stability in scores outside of this time frame.

Notably, the significant interaction between group and the linear effect of infant age showed that this decrease in language scores was reduced in the Reading Group, compared to the Control Group. While the two groups did not differ in Hearing and Language scores at 9 (main effect of group, with infant age centered at nine months: b(SE) = 2.202(1.634), *p* = 0.216 FDR corrected) and 12 (main effect of group, with infant age centered at 12 months: b(SE) = −1.144(1.743), *p* = 0.513 FDR corrected) months, the Reading Group showed significantly higher scores than the Control Group by the time they reached 18 (main effect of group, with infant age centered at 18 months: b(SE) = −6.343(2.588), *p* = 0.024 FDR corrected) and 24 (main effect of group, with infant age centered at 24 months: b(SE) = −10.324(3.758), *p* = 0.015 FDR corrected) months of age. This is especially notable, if considering that Reading Group infants started out with lower scores that the control group, at 3 (main effect of group, with infant age centered at 3 months: b(SE) = 10.971(2.555), *p* ≤ 0.001 FDR corrected) and 6 (main effect of group, with infant age centered at 6 months: b(SE) = 6.202(1.693), *p* = 0.001 FDR corrected) months of life.

To offer a clearer representation of this difference found across time between the two groups, change scores were calculated by subtracting, for each participant, Hearing and Language scores at time *t* from those found at time *t +* 1 (i.e., for each participant, subtracting the 3-month scores from the 6-month scores, subtracting the 6-month scores from the 9-months scores, etc.) (Table 2). Positive scores thus represented an increase at the specific time point compared to the previous one, while a negative score represented a decrease between time points, and a score of 0 indicated a stability between the two measurements. Visual inspection of plotted data for these change scores (see Figure 2) evidenced the presence of a quadratic trajectory.

A final growth curve model was thus tested, including the linear, quadratic effects of infant age and their interactions with group (see Table 4).

Significant main effects of linear, and quadratic infant age were found, with change scores that, regardless of group membership, started positive, but decreased consistently during the first nine months of life, reaching and maintaining negative values until the age of 18 months. After this age, when the infants reached 24 months, the last assessed age, their change scores (i.e., the difference in their score compared to their assessment at 18 months) returned to zero, indicating stability in the Hearing and Language scores between those two time points.

A significant effect of group was found showing that, throughout the investigated period, the Reading Group showed higher change scores, compared to the Control Group.

Among covariates, infant gender was found to significantly affect change scores (F(1, 84.80) = 10.06, *p* = 0.002), with female infants showing higher scores compared to male infants.

## 4. Discussion

The benefits of book-reading interventions on language development in full-term infants has been extensively investigated [63,64,65,66,67,68]. Conversely, prior to the present study, less attention has been devoted to the implementation of this intervention for preterm infants, despite the high risk for language delays that they face [8,9,10]. The present study therefore aimed to fill this gap, by longitudinally investigating the potential effects of a book-reading intervention, implemented in a Neonatal Intensive Care Unit (NICU), on preterm infant language development in the first 24 months of corrected age.

A first result regards the significant decrease of Hearing and Language quotients between 9 and 18 months of life. Notably, this effect emerged similarly in all infants, regardless of the presence of book-reading. Such a decrease may be related to the specific moment of assessment, as a similar trend was observed and reported by these preterm infants in a previous study [75], where scores for Extremely Low Birth Weight (ELBW) preterm infants significantly decreased from 9 to 12 months. In the previous study, this decrease was suggested as being brought about by an adjustment to new abilities acquired at the end of the first year, especially from a language standpoint, such as the use of symbolic gestures, long vocalizations, and lallations (the repetition of syllables, typical of the second half of the first year of infant life). The results of the present study would seem to confirm the previous findings, while also suggesting that the decrease observed in preterm infants may persist throughout the first half of the second year of life. Of further note, in a previous study [76], this decrease was only found in ELBW infants, with no effects found on Very Low Birth weight (VLBW) infants, a finding suggested to be explained by the greater severity of prematurity of ELBW versus VLBW infants. Conversely, in the present study the decrease emerged also for VLBW infants, suggesting that, while these infants might manage to successfully navigate the change in language skills at 12 months, difficulties might emerge at older ages (18 months), when the complexities of these skills increases further. Future studies which also consider the different levels of severity of prematurity might be able to test this hypothesis. 

The present results and those reported by Neri et al. [76] point towards the early identification of difficulties in preterm infants’ language skills, during the pre-lingual phase (from birth to 12 months of age) and at the beginning of the early-lingual phase (from 1 to 2.6 years of age), in addition to the assessment of children at older age [8,9,10]. Taken together, these findings would seem to support the need for an early assessment of infant language development in this population.

Interestingly, in the present study, no significant change in scores was found at 24 months of corrected age. This result was unexpected, due to the greater complexities of linguistic abilities at this age, and the many difficulties associated with mastering them, as evidenced by previous literature [8,9,10] showing that linguistic difficulties increase alongside child age. This greater complexity of language at 24 months is also reflected in the GMDS-R items, which, for example, require infants of this age to enunciate a specific number of words, to name objects appropriately, or to use combinations of words [79]. Future longitudinal studies would be needed to assess the trajectories of language development after the second year of infant life.

In line with our hypothesis, the decrease in language scores was found to be significantly lower in infants exposed to the book-reading intervention, who showed scores that were more stable and less inclined to decrease over time. This result, consistent with the literature on full term [63,64,65,66,67,68] and on preterm infants [69,70], supports the notion that book-reading interventions could be effective in improving cognitive development and preventing delays in language development in preterm infants.

Notably, however, while all previous studies on preterm children underline a positive effect of this intervention at the end of the second year of life, the present results could suggest a buffering effect against the decrease in language development that seems to start around the age of 9 months. Given the lack of studies on early book interventions in preterm infants, further studies are recommended in order to better explore the plausibility of our possible explanation.

Consideration should also be given to the infants’ age at the moment of intervention and assessment. Indeed, while the present study focused on neonates and their development in the first 24 months of corrected age, interventions in previous studies [69,70] were implemented when infants were older, with ages ranging 24–30 months. Present study findings may underline developmental stages differences, and suggest how book-reading interventions in younger infants may have a beneficial effect, not only through improving performance, but also through reducing the risk of onset of developmental difficulties. Indeed, the smaller reduction in scores seen in the Reading group suggests a beneficial effect, which seems to buffer the negative effects of preterm birth on language development.

Previous studies on book-reading interventions aimed at preterm infants [69,70], with their focus on older infants, were mainly focused on investigating the usefulness of this intervention for the treatment of language delays that were already present. The results reported here would instead support the use of book-reading interventions in a preventive perspective, supporting preterm infants and their families, before the onset of delays or impairments. From a more general standpoint, our findings, together with those by Braid and Bernstein [69] and Zuccarini and colleagues [70], support the implementation of book reading interventions in the first years of life as a fundamental tool for supporting language development in high-risk infants, such as preterm-born infants. 

Finally, despite it not being a specific aim of our study, we found a significant effect of infant gender on language outcomes, with higher scores in girls than in males. This result informs an open debate in literature, where several authors [84,87,88,89,90,91] consider gender to be strongly associated with verbal abilities, with others instead only suggesting a marginal effect [92,93,94]. The inconsistencies among studies are often associated with differences in infant age, with a limited number of studies recruiting preschool children [88,89,90] to investigate differences in emerging language skills. Our results could contribute to the literature on language skills in the first months of corrected age, supporting the need to consider infant gender, when studying possible interventions in support of language development.

### Limitations and Strengths

Although promising, the present study results should be considered as preliminary, and some limitations of the study should be noted. First, the limited sample size prevented the testing of more sophisticated hypotheses. Moreover, the recruited sample did not include a control group of full-term infants, which could have provided information on how intervention effects compared to those observed in normative samples. Future specific analyses that consider different groups of severity of prematurity (i.e., ELBW vs. VLBW) could be useful in furthering the understanding of the present results. 

Another limitation of this study relates to the absence of data about how and how often parents read to their infants after hospital discharge. If previous studies [69,70,73,74] have indeed shown that book-reading interventions are effective for improving parental attitudes towards reading at home, as well as improving how often parents read to their infants, an accurate measure of these dimensions would have strengthened our findings. The reported results should thus be considered cautiously, with future investigation recommended to consider these variables. 

Finally, future studies in this area would benefit from including measures of intervention effects on parental mental state and the quality of parent-infant interactions, as these may also be found to benefit from the intervention and may mediate the impact of the book-reading intervention on infant language development.

Notwithstanding the limitations above, this study also has several points of strength. Despite the empirical evidence reported on benefits of book-reading interventions for full-term infants, from the early months of life, this approach has rarely been applied in the context of prematurity, and only at later ages. Therefore, a strength of our study is the implementation of book-reading, starting in the stages of NICU hospitalization, a period where the exposure to parental voices is absent or mediated through taped registrations that lack the synchrony of a parents’ live presence. To our knowledge, no previous study has investigated the effects of a book-reading intervention during NICU hospitalization, making the present results promising and supporting the need for further investigations. Another element of strength is represented by the intensive longitudinal observations of infant language development, which enabled investigation of when the performance of premature infants started to become critical, and of when the benefits of the book-reading intervention began to be evident. Finally, a strength of this study regards the inclusion of VLBW infants, a population of preterm infants which has been recognized as being at high-risk for developmental delays, but which has not previously been included as specific target for this kind of intervention.

Future studies and replications are however needed to generalize results and support recommendations for clinical care.

## 5. Conclusions

The present study investigated the possible benefits of a book-reading intervention aimed at preterm infants, during their stay in the NICU, on their later language development. In summary, despite a change over time being observed in all infants, regardless of them having received the intervention or not, positive effects of the book-reading intervention were found. Specifically, when the change between assessments was positive, the improvement was higher in the reading group; at the same time, the change was found to be negative, the worsening of scores was smaller for those infants who had received the intervention. The specific and innovative aspect of our study regards the early implementation of the intervention (during NICU hospitalization). The study background is linked to previous literature which explores the benefits of early exposure to adult language in NICUs [25,26,27,36,48,49]. 

Despite being preliminary, our findings are in line with these previous studies, supporting the need to expose premature newborn infants to parental voices, to also reduce the risk of future delays in language development. This has considerable clinical implications, suggesting the use of book-reading interventions not only for the treatment of language delays, increasing the production of words and sentences, as supported by studies on late talkers [70], but also from a preventive perspective.

The implementation of book-reading interventions may also benefit the developmental care of preterm infants’ families. Indeed, this intervention may represent an important support for good parenting [95,96]. Preterm infants need affection, support, and attention, which calls for parents to offer full care to their infants, without limiting their responses to addressing physical needs. On the admittance in the NICU, however, parents may be confused, emotionally overwhelmed, and could react with unnatural coolness or detachment [97]. In this context, book-reading may help parents in positively engaging their infant and may help them cope with a difficult experience, reinstating their role as primary caregiver. The continued practice of book-reading may then become a “secure base” where comfort may be found, when NICU storms hit. Book-reading could represent a positive experience in the relationship with the infant and may be maintained after hospital discharge as well. 

Overall, the present study adds new findings to the literature, with potential clinical and practical implications for the care of preterm infants and their families. It is most probably never too early to start reading to children. 

## Figures and Tables

**Figure 1 ijerph-18-11361-f001:**
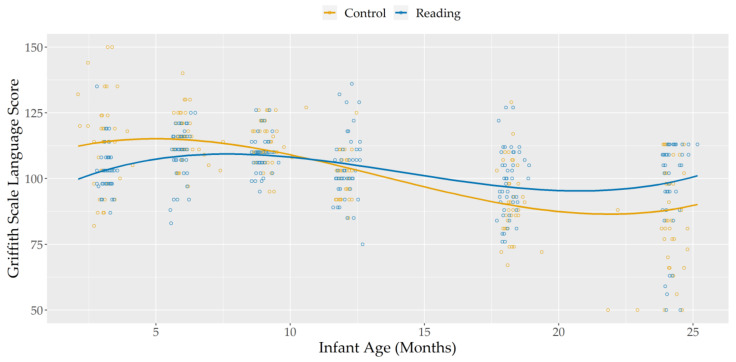
Griffiths Scale Language Scores, according to infant age and group.

**Figure 2 ijerph-18-11361-f002:**
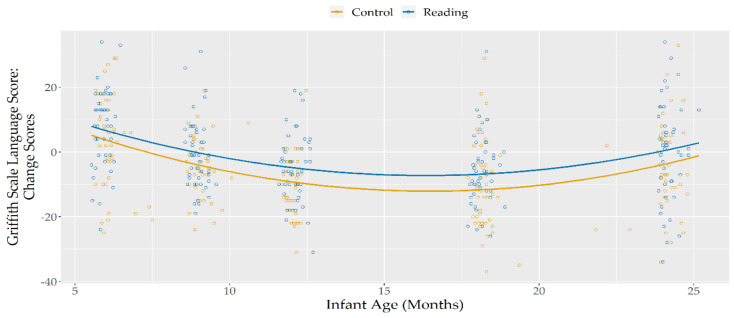
Hearing and Language Change Scores, according to infant age and group.

**Table 1 ijerph-18-11361-t001:** Main Sample Characteristics.

	Reading Group (N = 55)	Control Group (N = 45)	*p*
Infants			
Gender, % Male	56.4	62.2	0.698
Birth Weight, % ELBW	34.6	46.7	0.305
Gestational Age, % ELGA	47.3	60.0	0.286
Mothers			
Age (Yrs), M ± SD	35.20 ± 4.82	34.14 ± 5.10	0.301
Nationality, % Italian	85.5	91.1	0.578
Education			0.412
Middle School, %	16.4	20.0	
High School, %	47.3	60.0	
University, %	27.3	17.8	
Marital Status, % Married	49.1	64.4	0.128

Note. ELBW = Extremely Low Birth Weight; ELGA = Extremely Low Gestational Age.

**Table 2 ijerph-18-11361-t002:** Mean scores and changes scores at Hearing and Language Quotients according to infant age.

Infant Age	Reading Group (N = 55)		Control Group (N = 45)	
	M ± SD	Change Scores	M ± SD	Change Scores
3 months	102.50 ± 1.74	/	112.80 ± 1.93	/
6 months	109.48 ± 1.27	7.3	116.45 ± 1.41	3.66
9 months	108.41 ± 1.02	−0.64	111.75 ± 1.13	−4.70
12 months	103.20 ± 1.49	−5.19	101.75 ± 1.65	−10.00
18 months	97.09 ± 1.79	−6.27	91.75 ± 1.98	−10.0
24 months	97.11 ± 2.60	0.54	87.45 ± 2.88	−4.30

**Table 3 ijerph-18-11361-t003:** Growth Curve model effects on Hearing and Language Scores.

Effects	Degrees of Freedom	*F* Value	*p*
Group	1, 88.04	0.51	0.637
Infant Age (IA)	1, 285.77	135.06	≤0.001
IA (Quadratic)	1, 140.20	25.90	≤0.001
IA (Cubic)	1, 281.98	69.00	≤0.001
Group * IA	1, 285.77	9.42	0.004
Group * IA (Quadratic)	1, 140.21	1.44	0.348
Group * IA (Cubic)	1, 281.97	0.07	0.798

* FDR corrected, controlling for gestational age, infant birth weight, infant gender, and maternal education level.

**Table 4 ijerph-18-11361-t004:** Growth Curve model effects on Hearing and Language Change Scores.

Effects	Degrees of Freedom	*F* Value	*p*
Group	1, 88.32	5.40	0.038
Infant Age (IA)	1, 89.69	46.01	≤0.001
IA (Quadratic)	1, 89.63	51.18	≤0.001
Group * IA	1, 89.69	0.24	0.746
Group * IA (Quadratic)	1, 89.63	0.11	0.798

* FDR corrected, controlling for gestational age, infant birth weight, infant gender, and maternal education level.

## Data Availability

Data available on request due to privacy and ethical restrictions.

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
