# Peer review of "Parental Book-Reading to Preterm Born Infants in NICU: The Effects on Language Development in the First Two Years"

_ijerph, 2021, doi:10.3390/ijerph182111361_

Round 1

Reviewer 1 Report

IJERPH 1346515 Peer Review

Parental book-reading to preterm born infants in NICU: the effects on language development in the first 2 years

  • Overall, I agree with the authors that studying an intervention such as book reading during NICU hospitalization of VLBW infants is important. My concerns with the current manuscript include 1) English grammar and readability; 2) Lack of development of the background section to include more detailed information about the complexities of language development, known essential ingredients of language interventions and a clear rational for the use of this book reading intervention in the NICU.
  • Some specific notes:
  • Language editing for flow and readability throughout. Many sentences are difficult to follow, long or incorrect grammar. For example: Pg 2 lines 43-47; pg 2 lines 47-49; pg 2 lines 63-64.
  • Pg 2 line 48-49, just need “NICU” here not the full name, you have already established the acronym in line 42.
  • Citation for pg 2 line 86-87 regarding parents feeling awkward or embarrassed.
  • Overuse of the word “nevertheless” throughout.
  • Please be consistent in the use of “preterm infants” or “preterm babies” throughout, not both.
  • Pg 3 line 131, you have called the intervention here the “tale reading interventions” but prior called it a “book reading intervention” choose one and be consistent throughout.
  • Background –
    • As the authors have stated, it is important to investigate early interventions for VLBW infants that can protect or prevent later language delays. However, in this article there is not enough emphasis in the background on the complexities of preterm birth and the relationship to later development. More detailed information about the NIDCAP specific recommendations regarding “appropriate amounts and kinds of stimulation” would help here.
    • Missing information in the background regarding WHAT the book reading intervention provides that is essential to the preterm infant experience in the NICU. What is the mechanism that you expect to cause the change in language development? Is it exposure to the sounds of parental voice? Is it exposure to words? Is it exposure to “infant-directed” speech sounds? Please revise the background to explore deeper the auditory experience that book reading would provide and what you believe the mechanism of change to be in this intervention. Why do you believe that reading will be associated with higher language abilities for this population?
  • Materials and methods
    • Please revise to provide additional details about the book reading intervention and answer the following questions:
      • Was the same book provided to the families with the exception of the non-Italian speaking families?
      • What considerations or characteristics were used in deciding the book choices?
      • Who provided information to the parents regarding book reading? Was the information provided to all families the same? What fidelity measures were in place to ensure each family received the same information?
      • How were the NICU staff trained to support parents in reading? Again, what fidelity measures were in place to ensure each NICU staff received the same training and was implementing the information to the families in the same way?
      • How was information provided to the non-Italian speaking families?
      • Was any data collected on how often or for how long the families read to their infant in the NICU and after discharge? Without data collected on if/and for how long the families read, the intervention would be more accurately be described as a parent information intervention on the benefits of reading, and NOT a reading intervention because you do not know if parents actually read to their infant, only that they were provided information and resources for reading.
    • Any of these questions that cannot be directly accounted for in this study should be included in the limitations and the study revised to more accurately represent what IS known about the intervention that was provided. This will also impact interpretation of the results and discussion.

Author Response

We sincerly thank reviewer for all suggestions that help us to improve the manuscripts. All revision and comments are in attached files.

Reviewer 2 Report

Dear authors,

find my comments and suggestions attached.

best wishes and hoping to see a new version

Author Response

We sincerly thank reviewer fro all suggestions that help us to improve the manuscript. Please see the attachment for comments and responces.

Round 2

Reviewer 1 Report

Thank you for your extensive revisions to the manuscript. It is now a much stronger presentation of your important work.

Author Response

We thank reviewer. We further check English language and made corrections

Reviewer 2 Report

The paper in its current form is much better than the last. However, I still have some concerns and suggestions as you will find in the attached pdf

best

Author Response

We thank reviewer. All manuscript was revised according with suggestions.

Red sentences are our comments to revievwer.
